# Emotional State of Parents of Children Diagnosed with Cancer: Examining Religious and Meaning-Focused Coping

**Natalia Ziółkowska [1], Kamilla Bargiel-Matusiewicz [1,2,]\* and Ewa Gruszczyńska [1]**

[1] Institute of Psychology, SWPS University of Social Sciences and Humanities, Chodakowska Street 19/31, 03-815 Warsaw, Poland; nziolkowska@st.swps.edu.pl (N.Z.); egruszczynska@swps.edu.pl (E.G.)

[2] Faculty of Psychology, University of Warsaw, Stawki Street 5/7, 00-183 Warsaw, Poland

\* Correspondence: kmatusiewicz@psych.uw.edu.pl

**Abstract:** For parents, a child's oncological disease is a critical life event with a high burdening potential, which changes the functioning of the whole family on many different levels. It triggers various coping strategies with this situation, including religious-based coping. This topic has been somewhat rarely explored, and thus, the aim of the study was to examine the relationship between the emotional state and religious and meaning-focused coping among parents of children diagnosed with cancer. A total of 147 parents participated in this study. Hierarchical regression analysis showed that correlates of positive emotions in the studied group were the economic situation, the time from diagnosis, positive reappraisal and negative religious coping. Only one significant correlate of negative emotions was identified. There is some support for the incremental validity of negative religious coping in relation to meaning-focused coping.

**Keywords:** religious coping; emotions; parents; children with cancer

## 1. Introduction

A child's oncological disease is a critical life event with the high burdening potential for most parents (Gouvela et al. 2017). Each year in the United States of America, nearly 12,500 new oncological diseases are diagnosed among children and adolescents (National Cancer Institute 2013; Kazak and Noll 2015), which is the second-highest cause of death among children (Kazak and Noll 2015). Such a diagnosis causes distress and disrupts the functioning of the whole family, especially the parents (Gage-Bouchard et al. 2013; Ljungman et al. 2014). The literature focuses on the process of mourning for a healthy child and adapting to the situation caused by a child's disease (Cantwell-Bartl 2018; Dorfmuller and Dietzfelbinger 2011; McCarthy et al. 2012). Oncological treatment is often accompanied by anxiety related to the short-term outcomes, side effects, and the likelihood of recurrence (Santos et al. 2018). Thus, a life-threatening somatic disease in a child is a source of chronic stress among parents and triggers various coping strategies.

Meaning-focused coping, introduced by Folkman (1997), may be applicable in this context. This concept is a result of the modification of the traditional stress and coping model (Lazarus and Folkman 1984), which assumes that cognitive appraisal of an imbalance between perceived external or internal demands and the perceived personal and social resources is crucial for stress identification, emotions experienced, and activation of the coping process. This leads to a cognitive–motivational–relational theory to explain and predict emotions, with a core relational theme distinctive for each emotion (Lazarus 2000). Although the theory is currently regarded as disputable (Moors and Scherer 2013), it corresponds to the importance of the meaning-making processes. In contrast to global meaning,

situational meaning refers to a particular context (Park 2010) and searching for it can be an essential part of coping with long-lasting and low controllable situations (Folkman and Moskowitz 2007; Park and Folkman 1997). Thus, meaning-focused coping is defined as "appraisal-based coping in which the person draws on his or her beliefs (e.g., religious, spiritual, or beliefs about justice), values (e.g., mattering), and existential goals (e.g., purpose in life or guiding principles) to motivate and sustain coping and well-being during a difficult time" (Folkman 2008, p. 5).

Folkman (1997); Folkman and Moskowitz (2000) emphasized the role of meaning-focused coping in sustaining and activating positive emotions in the context of chronic stress. These include positive reappraisal, revision of goals, and related coping, as well as creating positive events and enriching ordinary events with positive meaning (Folkman 1997). More precisely, positive reappraisal is a cognitive process during which the stressful situation is reconstructed as valuable and beneficial, thus, it is perceived as a possibility for personal development and gaining more in-depth resources, just like wisdom or higher quality relations with others. Goal revision and adjusting problem-focused coping to the current demands of the situation enables people to preserve a sense of control and agency, especially as a disease is rarely a controllable source of stress. Finally, proactive planning for events inducing positive affective experiences and infusing ordinary events with special meaning leads to positive emotions that provide a break from the chronic strain and allows people to restore resources necessary to continue coping (Folkman 2008; Folkman and Moskowitz 2000).

Similarly, religious coping is recognized as playing a significant role in uncontrollably stressful situations (Paloutzian and Park 2015). Pargament (1997, p. 32) described religion as "a search for significance in ways related to the sacred" and religious coping as "efforts to understand and deal with life stressors in ways related to the sacred" (Pargament et al. 2011, p. 52). Pargament et al. (2000) underlined that religious coping has five functions during difficult events: searching for meaning, control, comfort/spirituality, intimacy/spirituality, and life transformation. Although acting on the same basis, two forms of religious coping can be differentiated, i.e., positive and negative strategies. Among positive religious coping, there is, for example, seeking spiritual support, benevolent religious reappraisal, or spiritual connection. Negative religious coping includes punishing god reappraisals, demonic reappraisal, or spiritual discontent (Pargament et al. 2011; Pargament et al. 1998; Davis and Kiang 2018). Thus, positive and negative religious coping are regarded as distinct phenomena, which, although potentially interrelated, are not only opposite poles of the same dimension. This theoretical assumption was confirmed by both a two-factor structure of the measurement tool and different correlations with measures of physical and mental health (Pargament et al. 2011). Specifically, the general patterns of findings show positive correlations between positive religious coping, well-being, and quality of life (Abu-Raiya et al. 2011; Gall et al. 2009; Khanjari et al. 2018). Meanwhile, for negative religious coping, the opposite or no significant relationship has been reported (Abu-Raiya et al. 2015; Gall et al. 2009; Vitorino et al. 2018). The results showed that religious coping is triggered early in the adaptation process to the diagnosis of cancer and following treatment (Gall et al. 2009). However, as already mentioned, studies among parents of children suffering from the serious somatic disease are limited.

As such, turning to religion is a broader concept that describes the coping strategy understood as seeking comfort, support, and/or guidance from the divine (Carver et al. 1989). Thus, it may play different functions in the coping process, resulting in a wide range of adaptational outcomes (Ano and Vasconcelles 2005). For instance, in exploratory factor analysis, turning to religion was reported to be a distinct factor (Carver et al. 1989) not loaded sufficiently on any factor, but loaded on one factor with using emotional support and instrumental support (Schottenbauer et al. 2006) or with positive reframing and humor (Kellezi et al. 2009). The meta-analysis by Krägeloh (2011) revealed that the results are dependent on the level of the analysis. At the subscale (i.e., coping indicators) level, turning to religion shares a common variance with active coping and positive reframing, whereas at the item level, it is related more to strategies like denial or mental disengagement. Being unspecific

and context-sensitive may cover other methods of religious coping than those specified by Pargament (Pargament et al. 2011; Pargament et al. 1998).

Although meaning-making processes can be influenced to a large extent by religion (Park 2010, 2013), they are different constructs, as well as meanings, that can be searched for within a given religion as well as outside any religious system (Schnell and Keenan 2013; Van Uden and Zondag 2016). Religious coping was also observed in secular societies (Pedersen et al. 2013), suggesting that it may be, only to a sudden or partial extent, related to declared religiosity (Schnell 2012). Thus, the question remains if religious coping is a special case of meaning-focused coping. This issue may be analyzed by examining whether religious coping demonstrates significant incremental variance beyond meaning-focused coping when explaining the emotional state of parents of children with oncological diseases.

## 2. Materials and Methods

### 2.1. Study Aim

The main aim of this study was to analyze religious coping strategies and positive and negative emotions among parents of children suffering from an oncological disease. The secondary aim was to examine the relationship between religious coping and non-religious meaning-focused coping. Specifically, positive reappraisal was considered as it refers directly to a change in the meaning of the situation, which is also a core of religious coping. Thus, both these methods of coping are cognitive in nature and based on appraisal-related processes.

The two main research questions asked in this study were:

1. Is the emotional state of the parents related to religious coping as well as to meaning-focused coping (after controlling for basic sociodemographic and clinical variables)?
2. Is there a significant incremental variance in religious coping beyond meaning-focused coping that explains the emotional state of the parents?

We hypothesized that positive emotions are positively correlated with positive religious coping and with meaning-focused coping, but negatively with negative religious coping. Conversely, negative emotions are positively associated with negative religious coping and negatively with both positive religious coping and meaning-focused coping. Finally, we assumed that religious coping after controlling for meaning-focused coping still significantly adds to the explained variance of the positive and negative emotions of parents of children with oncological diseases.

### 2.2. Procedure and Participants

The study was conducted in the specialized children oncology department at two high reference public medical centers: Department of Oncology of the Children's Memorial Health Institute in Warsaw and the Children Hematology and Oncology Ward of Prof. S. Szyszko Medical University of Silesia's Clinical Hospital No 1 in Katowice. The study received approval and a permit from the institutional ethics committee (opinion no. 5/2019). The participants included 147 parents of children diagnosed with an oncological disease remaining in active treatment in oncology clinics, 78.9% of whom were mothers and 21.1% were fathers. The average age of the parents 36.80 years (SD = 6.53 years). The majority of the parents had higher (53.8%) or secondary (31.7%) education. Most of the parents were married or in a domestic partnership (92.5%). In most cases, only one parent took care of the child in the hospital (73.5%). The parents declared being mainly Catholic (91.1%). Most of the children were boys (60.5%). The average age of the children was 7.89 years (SD = 5.53 years), and at the beginning of the illness, they were 6.17 years (SD = 4.98 years). Most of the children were diagnosed with an oncological disease for the first time, with the recurrence rate for the sample being 12.2%. Tumors were mostly solid (57.1%), hematopoietic and lymphatic system tumors constituted 39.5%. Most of the children were hospitalized (61.8%) and undergoing chemotherapy (68.7%) at the time of the study.

*2.3. Measures*

Three questionnaires were used in the study. The Brief Scale of Religious Coping (Brief RCOPE, Pargament 1997, 2011, adaptation by Jarosz (2011) was applied to measure religious coping. RCOPE has been successfully used in studies among diversified populations (Abu-Raiya and Pargament 2015). Brief RCOPE has good psychometric characteristics (Pargament et al. 2011). Some controversies have been expressed regarding its validity among secular societies (Alma et al. 2003), which, however, is not the case in this study. The scale consists of 14 items describing positive and negative methods of religious coping in stressful situations, accompanied by a 5-point scale ranging from 1 = "definitely not" to "definitely yes". In the instructions, parents were asked to refer directly to their coping with the ongoing stressful situation of their child's disease. Two indicators were obtained, positive and negative religious coping, each by summing answers to the relevant seven items. Higher values point to a higher intensity of given coping strategies. In the current study, Cronbach's $\alpha$ values were 0.95 and 0.91 for positive and negative religious coping, respectively.

The emotional state of the parents during the last 4 weeks was assessed with 15 adjectives based on questionnaire proposed by Folkman and Lazarus (1985). The participants marked their answers on a 5-point scale (from 0 = "not at all" to 4 = "very strongly"). Two indicators were obtained by summing the answers for positive (hopeful, eager, happy, pleased, relieved, exhilarated, optimistic) and negative emotions (worried, anxious, angry, sad, disappointed, insecure, helpless, bored). In our sample, Cronbach's $\alpha$ values for the positive and negative emotions subscales were 0.81 and 0.83, respectively.

Finally, two subscales from Brief RCOPE (Carver et al. 1989, adaptation by Juczyński and Ogińska Bulik (Juczyński 2009) were used to evaluate positive reappraisal and turning to religion. Each subscale consists of two items. The answers were provided on a 4-point scale (from 0 = "almost never do that" to 3 = "almost always do that"), then summed. For turning to religion, Cronbach's $\alpha$ was 0.89, whereas, for positive reappraisal, it was 0.61, which suggests questionable internal consistency of this subscale. It will be discussed as a limitation of the study since this result appears to be a sample-related effect.

Additionally, information concerning a child's disease and treatment as well as their parent's sociodemographic characteristics were collected.

## 3. Results

*3.1. Descriptive Statistics and Correlational Analysis*

Descriptive statistics and correlations among the studied variables are provided in Table 1. The percentage of missing data was low (from 0% to 5.4%). Positive emotions correlated negatively with negative coping and positively with positive reappraisal. However, both these correlations were weak. Negative emotions correlated significantly only with negative religious coping. To some degree, higher positive religious coping co-occurred with higher negative religious coping. Positive reappraisal was unrelated to negative religious coping and only weakly related to positive religious coping. The strongest relationship was obtained for turning to religion and positive religious coping (about 64% of shared variance), whereas the same relationship for negative religious coping did not differ significantly from zero.

**Table 1.** Descriptive statistics and correlations among the studied variables (*N* = 147).

| Variable | M | SD | Skewness | Kurtosis | Min | Max | 1 | 2 | 3 | 4 | 5 |
|---|---|---|---|---|---|---|---|---|---|---|---|
| Positive emotions | 6.29 | 3.75 | 0.37 | −0.42 | 0 | 17 | 1 | | | | |
| Negative emotions | 9.86 | 4.68 | 0.13 | −0.67 | 0 | 20 | −0.53 ** | 1 | | | |
| Positive religious coping | 22.38 | 8.14 | −0.48 | −0.81 | 7 | 35 | 0.09 | 0.06 | 1 | | |
| Negative religious coping | 17.01 | 7.41 | 0.32 | −0.72 | 7 | 35 | −0.22 ** | 0.33 ** | 0.30 ** | 1 | |
| Positive reappraisal | 1.69 | 0.76 | −0.26 | −0.28 | 0 | 3 | 0.29 ** | −0.13 | 0.20 * | −0.04 | 1 |
| Turning to religion | 1.59 | 1.11 | −0.12 | −1.38 | 0 | 3 | 0.01 | 0.12 | 0.81 ** | 0.16 | 0.14 |

Note: M = Mean, SD = Standard deviation, Min = Minimum, Max = Maximum, ** $p < 0.01$, * $p < 0.05$.

### 3.2. Hierarchical Regression Analysis

Hierarchical regression was used to address both research questions. In the first step, a parent's characteristics were included, followed by the basic sociodemographic and clinical description of the child in the second and third steps, respectively. In the fourth step, we introduced positive reappraisal and tuning to religion, which can be regarded as more general meaning-focused coping. Finally, the religious coping divided into positive and negative modes was added. The analyses were conducted separately for positive and negative emotions as explained variables. Collinearity was checked using a variance inflation factor (VIF), which ranged from 1.07 to 5.91.

For positive emotions, the model including meaning-focused coping surpassed the threshold of statistical significance, explaining 15% of the variance of positive emotions, with positive reappraisal as the main contributor. Adding religious coping in the final step led to an increase in the explained variance of 5% ($p = 0.02$). Thus, the resultant model identified four correlates of positive emotions after controlling for other studied variables: economic situation, time since diagnosis (the longer, the higher the positive emotions), positive reappraisal (higher values relate to more positive emotions), and negative religious coping (higher values relate to lower positive emotions). For positive emotions, negative religious coping showed an incremental variance with regards to positive reappraisal and turning to religion. Results are presented in Table 2.

**Table 2.** Results of hierarchical regression analysis for positive emotions.

| | Model 1 | | | Model 2 | | | Model 3 | | | Model 4 | | | Model 5 | | |
|---|---|---|---|---|---|---|---|---|---|---|---|---|---|---|---|
| | B | SE B | Beta | B | SE B | Beta | B | SE B | Beta | B | SE B | Beta | B | SE B | Beta |
| **Parent's characteristics** | | | | | | | | | | | | | | | |
| Sex | 0.99 | 0.89 | 0.11 | 0.87 | 0.91 | 0.09 | 1.33 | 0.92 | 0.14 | 1.85 | 0.89 | 0.20 * | 1.67 | 0.87 | 0.18 |
| Age | −0.04 | 0.05 | −0.07 | −0.04 | 0.07 | −0.07 | −0.04 | 0.07 | −0.07 | −0.08 | 0.07 | −0.14 | −0.09 | 0.07 | −0.16 |
| Education | −0.63 | 0.70 | −0.08 | −0.57 | 0.73 | −0.08 | −0.37 | 0.74 | −0.05 | 0.02 | 0.70 | 0.00 | −0.13 | 0.69 | −0.02 |
| Relationship status | −0.91 | 1.42 | −0.06 | −0.70 | 1.45 | −0.05 | −1.29 | 1.47 | −0.08 | −1.15 | 1.38 | −0.07 | −1.82 | 1.37 | −0.12 |
| Economic situation | 1.01 | 0.59 | 0.16 | 0.94 | 0.60 | 0.14 | 1.20 | 0.60 | 0.19 * | 1.14 | 0.56 | 0.18 * | 1.14 | 0.55 | 0.18 * |
| Professional work | 0.44 | 0.81 | 0.05 | 0.35 | 0.81 | 0.04 | 0.33 | 0.82 | 0.04 | 0.39 | 0.77 | 0.05 | 0.62 | 0.76 | 0.08 |
| Hospital care (one parent or parents) | 0.12 | 0.78 | 0.01 | −0.04 | 0.79 | −0.01 | −0.07 | 0.80 | −0.01 | −0.06 | 0.77 | −0.01 | −0.27 | 0.75 | −0.03 |
| **Child's characteristics** | | | | | | | | | | | | | | | |
| Sex | | | | 0.67 | 0.71 | 0.09 | 0.61 | 0.73 | 0.08 | 0.27 | 0.70 | 0.03 | 0.69 | 0.70 | 0.09 |
| Age at the time of diagnosis | | | | −0.14 | 0.12 | −0.18 | −0.12 | 0.15 | −0.16 | −0.06 | 0.15 | −0.08 | 0.01 | 0.15 | 0.01 |
| Age now | | | | 0.12 | 0.11 | 0.17 | 0.14 | 0.14 | 0.17 | 0.09 | 0.13 | 0.13 | 0.02 | 0.13 | 0.03 |
| Siblings | | | | −0.92 | 0.84 | −0.11 | −0.77 | 0.85 | −0.09 | −0.94 | 0.81 | −0.11 | −0.40 | 0.81 | −0.05 |
| Type of cancer | | | | | | | 0.00 | 0.00 | 0.04 | 0.00 | 0.00 | 0.05 | 0.00 | 0.00 | 0.07 |
| Time from diagnosis (months) | | | | | | | 0.03 | 0.02 | 0.22 | 0.03 | 0.02 | 0.24 | 0.04 | 0.02 | 0.28 * |
| Recurrence | | | | | | | 1.53 | 1.22 | 0.14 | 0.96 | 1.17 | 0.09 | 0.85 | 1.14 | 0.08 |
| Hospitalization | | | | | | | −0.34 | 0.83 | −0.04 | −0.32 | 0.78 | −0.04 | −0.28 | 0.76 | −0.04 |
| Preparation for surgery | | | | | | | 1.66 | 1.48 | 0.11 | 1.60 | 1.40 | 0.10 | 1.63 | 1.36 | 0.11 |
| Preparation for transplant | | | | | | | 2.43 | 1.36 | 0.19 | 2.43 | 1.28 | 0.19 | 1.45 | 1.29 | 0.11 |
| Chemotherapy | | | | | | | 0.62 | 1.14 | 0.08 | 0.22 | 1.09 | 0.03 | 0.25 | 1.07 | 0.03 |
| Radiotherapy | | | | | | | 0.70 | 1.37 | 0.05 | 0.95 | 1.30 | 0.06 | 1.44 | 1.28 | 0.10 |
| Supportive treatment | | | | | | | 1.37 | 1.18 | 0.14 | 1.38 | 1.11 | 0.14 | 1.18 | 1.09 | 0.12 |
| **Meaning-focused coping** | | | | | | | | | | | | | | | |
| Positive reappraisal | | | | | | | | | | 1.56 | 0.44 | 0.32 * | 1.40 | 0.44 | 0.29 * |
| Turning to religion | | | | | | | | | | 0.32 | 0.31 | 0.10 | 0.00 | 0.52 | 0.00 |
| **Religious coping** | | | | | | | | | | | | | | | |
| Positive religious coping | | | | | | | | | | | | | 0.07 | .07 | 0.16 |
| Negative religious coping | | | | | | | | | | | | | −0.14 | .05 | −0.28 * |
| $R^2$, adjusted $R^2$ | 0.05; | | | 0.28; | | | 0.19; | | | 0.30; | | | 0.35; | | |
| | 0.00 | | | −0.01 | | | 0.04 | | | 0.15 | | | 0.20 | | |
| F | 0.93 | | | 0.81 | | | 1.65 | | | 7.97 | | | 3.93 | | |
| $\Delta R^2$ | 0.05 | | | 0.03 | | | 0.11 | | | 0.11 | | | 0.05 | | |
| $p$ for $\Delta R^2$ | 0.48 | | | 0.52 | | | 0.11 | | | 0.001 | | | 0.02 | | |

Note: * $p < 0.05$.

For negative emotions, in the final step, negative religious coping was identified as the only significant correlate, which resulted in an increase in the explained variance of 6% ($p = 0.02$). Thus, as assumed, higher negative religious coping was related to higher negative emotions after controlling for all the other variables, which demonstrated its incremental variance. Results are presented in Table 3.

**Table 3.** Results of hierarchical regression analysis for negative emotions.

| | Model 1 | | | Model 2 | | | Model 3 | | | Model 4 | | | Model 5 | | |
|---|---|---|---|---|---|---|---|---|---|---|---|---|---|---|---|
| | B | SE B | Beta | B | SE B | Beta | B | SE B | Beta | B | SE B | Beta | B | SE B | Beta |
| **Parent's characteristics** | | | | | | | | | | | | | | | |
| Sex | −1.52 | 1.08 | −0.13 | −1.45 | 1.13 | −0.13 | −1.91 | 1.17 | −0.17 | −1.84 | 1.18 | −0.16 | −1.65 | 1.16 | −0.14 |
| Age | −0.09 | 0.07 | −0.13 | −0.08 | 0.08 | −0.12 | −0.07 | 0.09 | −0.10 | −0.07 | 0.09 | −0.11 | −0.06 | 0.09 | −0.08 |
| Education | 0.86 | 0.85 | 0.09 | 0.84 | 0.90 | 0.09 | 0.86 | 0.93 | 0.09 | 0.53 | 0.93 | 0.06 | 0.72 | 0.91 | 0.08 |
| Relationship status | −0.04 | 1.74 | 0.00 | −0.34 | 1.78 | −0.02 | 0.21 | 1.86 | 0.01 | 0.07 | 1.84 | 0.00 | 0.96 | 1.81 | 0.05 |
| Economic situation | −0.51 | 0.72 | −0.06 | −0.44 | 0.74 | −0.06 | −0.73 | 0.76 | −0.09 | −0.69 | 0.75 | −0.09 | −0.68 | 0.73 | −0.09 |
| Professional work | −0.36 | 0.98 | −0.04 | −0.26 | 1.00 | −0.03 | −0.13 | 1.03 | −0.01 | −0.14 | 1.02 | −0.01 | −0.48 | 1.00 | −0.05 |
| Hospital care (one parent or parents) | −0.55 | 0.95 | −0.05 | −0.46 | 0.97 | −0.05 | −0.45 | 1.01 | −0.04 | −0.05 | 1.02 | −0.01 | 0.25 | 1.00 | 0.02 |
| **Child's characteristics** | | | | | | | | | | | | | | | |
| Sex | | | | −0.21 | 0.88 | −0.02 | −0.02 | 0.93 | 0.00 | −0.06 | 0.93 | −0.01 | −0.64 | 0.93 | −0.07 |
| Age at the time of diagnosis | | | | 0.13 | 0.15 | 0.14 | 0.18 | 0.20 | 0.19 | 0.10 | 0.20 | 0.10 | 0.00 | 0.19 | 0.00 |
| Age now | | | | −0.12 | 0.14 | −0.14 | −0.18 | 0.17 | −0.22 | −0.13 | 0.17 | −0.16 | −0.05 | 0.17 | −0.06 |
| Siblings | | | | 1.12 | 1.04 | 0.11 | 0.88 | 1.08 | 0.08 | 1.23 | 1.08 | 0.12 | 0.49 | 1.08 | 0.05 |
| Type of cancer | | | | | | | 0.00 | 0.00 | 0.07 | 0.00 | 0.00 | 0.07 | 0.00 | 0.00 | 0.05 |
| Time from diagnosis (months) | | | | | | | −0.01 | 0.02 | −0.09 | −0.01 | 0.02 | −0.07 | −0.02 | 0.02 | −0.12 |
| Recurrence | | | | | | | −1.57 | 1.54 | −0.12 | −0.83 | 1.56 | −0.06 | −0.68 | 1.51 | −0.05 |
| Hospitalization | | | | | | | 0.08 | 1.05 | 0.01 | −0.10 | 1.04 | −0.01 | −0.15 | 1.01 | −0.02 |
| Preparation for surgery | | | | | | | −1.30 | 1.86 | −0.07 | −0.84 | 1.86 | −0.04 | −0.87 | 1.81 | −0.05 |
| Preparation for transplant | | | | | | | −2.75 | 1.71 | −0.18 | −2.58 | 1.70 | −0.16 | −1.21 | 1.71 | −0.08 |
| Chemotherapy | | | | | | | −0.91 | 1.45 | −0.09 | −0.41 | 1.45 | −0.04 | −0.39 | 1.42 | −0.04 |
| Radiotherapy | | | | | | | −1.19 | 1.73 | −0.07 | −0.94 | 1.73 | −0.05 | −1.63 | 1.69 | −0.09 |
| Supportive treatment | | | | | | | −0.92 | 1.49 | −0.08 | −0.85 | 1.47 | −0.07 | −0.54 | 1.44 | −0.04 |
| **Meaning-focused coping** | | | | | | | | | | | | | | | |
| Positive reappraisal | | | | | | | | | | −1.10 | 0.58 | −0.18 | −0.94 | 0.59 | −0.16 |
| Turning to religion | | | | | | | | | | 0.57 | 0.41 | 0.14 | 0.82 | 0.68 | 0.20 |
| **Religious coping** | | | | | | | | | | | | | | | |
| Positive religious coping | | | | | | | | | | | | | −0.07 | 0.09 | −0.13 |
| Negative religious coping | | | | | | | | | | | | | 0.19 | 0.07 | 0.31 ** |
| $R^2$, adjusted $R^2$ | 0.05; −0.01 | | | 0.06; −0.03 | | | 0.13; −0.04 | | | 0.16; −0.01 | | | 0.23; 0.05 | | |
| F | 0.83 | | | 0.44 | | | 0.93 | | | 2.3 | | | 4.16 | | |
| $\Delta R^2$ | 0.05 | | | 0.01 | | | 0.07 | | | 0.04 | | | 0.06 | | |
| p for $\Delta R^2$ | 0.57 | | | 0.78 | | | 0.51 | | | 0.11 | | | 0.02 | | |

Note. ** $p < 0.01$.

## 4. Discussion

The aim of this study was to analyze the relationship between emotional state and religious and meaning-focused coping among parents of children diagnosed with an oncological disease. Two hypotheses were formulated. Positive emotions were assumed to be explained by higher positive religious coping and meaning-focused coping and lower negative religious coping. For negative emotions, the reverse patterns of relationships were expected.

The preliminary correlational analysis showed that there are two significant correlations for positive emotions: negative with negative religious coping and positive with positive reappraisal. For negative emotions, only a positive relationship with negative religious coping was noted. Positive religious coping shared some similarities with meaning-focused coping, especially turning to religion. Thus, in regression analysis, the unique contribution of each coping strategy was tested after adjusting for other important contextual variables. The pattern stayed the same. Thus, our hypotheses were confirmed only partially. Positive emotions of parents who cope with an oncological disease of the child were related to positive reappraisal and negative religious coping, without a significant effect for positive religious coping. Of all the variables included in the model, the parents' negative emotions were only weakly explained by negative religious coping. Thus, negative religious coping may affect both valences, whereas positive reappraisal seems more specific for positive emotions.

Incremental validity of negative religious coping in explaining emotions could be regarded as supported empirically in our study, i.e., it added significant explanatory power for both positive and negative emotions over variables previously included in the models. No such effect was noted for positive religious coping. Tarakeshwar and Pargament (2001) noted that positive religious coping was unrelated to anxiety and depression among parents of children with autism, but was positively associated with stress-related growth, whereas negative religious coping was correlated only with depression. In another study, neither positive nor negative religious coping strategies were significantly related to the well-being of mothers of children with autism (Davis and Kiang 2018). Finally, Vitorino et al. (2018) observed that negative religious coping was associated with depression among family caregivers of pediatric cancer patients, also after adjusting for sociodemographic variables and positive

religious coping, which was unrelated to depression. However, positive reappraisal was not included in any of these studies. Thus, the question as to what degree positive religious coping can be a specific form of positive reappraisal is important both theoretically and empirically. In this context, negative religious coping can be regarded as a different construct, although with a more pronounced effect on negative emotions.

According to Folkman's conceptualization (Folkman 1997, 2008), meaning-focused coping positively relates to the well-being of a person by sustaining positive affective states under stress, and its core is looking for positive changes in the appraisal of the situation. However, the change in meaning to more negative can also occur in the process of dealing with stress. In a framework of the meaning-making model (Park 2010; Park and Folkman 1997), such contextual negative changes are the effect of a lack of integration between situational and global meaning, which is a failure of effective coping to reduce this stress-evoked mismatch. This could be called "negative reappraisal". Another condition occurs when more fundamental beliefs are affected and transformed due to such changes, leading to more negative global meaning. The item content of strategies under the scope of negative religious coping indicates that they may play a role in enhancing the negative meaning of the situation, as they involve and sustain negative attributions of a difficult situation as god's punishment, abandoning, and lack of mercy (Pargament et al. 2011; Pargament et al. 1998). However, using such strategies is rather the effect than a cause of global meaning, but it cannot be excluded that it could lead to further negativity of core beliefs.

Our findings highlight that negative religious coping may foster less positive emotions and more negative emotions among caregiving parents, which potentially makes them more prone to developing depressive symptoms (Folkman 1997; Pargament et al. 2004). It can be hypothesized that negative reappraisal is a crucial explanatory mechanism for this process as an equivalent of constant searching for meaning without finding it (Davis and Novoa 2013). It is therefore surprising that so little attention has been paid to negative reappraisal as a possible method of coping with stress, although, to some extent, this negative meaning-making is likely to be covered under studies of rumination (Nolen-Hoeksema et al. 2008). This raises a new question as to the extent to which negative religious coping is a specific form of ruminating.

However, this study is not without limitations. First of all, this study was cross-sectional in nature, which prevents the formulation of conclusions regarding causality. Therefore, in future studies, despite the difficulties connected with the availability of the study group and the study being time-consuming, a longitudinal study would be worth conducting. Although the sample size was larger than in other studies of the parents of children diagnosed with chronic illness reported in this paper, the findings may have insufficient statistical power. Given the low number of fathers (in comparison to mothers) in the sample, the findings may be biased toward this group of caregivers. Cronbach's α for positive reappraisal scale was below satisfactory, which may be a sample-related effect. In other studies using this subscale, the reported reliability measured using the same indicator was usually higher (Perczek et al. 2000). A closer look at the content of the items (i.e., I have been trying to see it in a different light, to make it seem more positive, I've been looking for something good in what is happening) suggests that the second item may have different meanings for parents of children with possibly fatal somatic conditions. First, the response may be an act of denial more than positive reframing. Second, as directed toward the future, parents may feel very insecure in such context, so it may be perceived as not valid or even increasing negative emotions. This issue needs further research.

All the effects observed in our study should be treated with caution as they are weak and affected by a large number of sociodemographic and clinical variables without any relationships with the explained variables, which can be regarded as a result in itself. This suggests that when the sample is relatively homogenous as to a child's diagnosis, the objective characteristics of the situation may differentiate the emotional response of the parents to a limited degree only. Therefore, the role of giving meaning to a situation becomes crucial, with coping fueled by religious beliefs as an important part of this process, with a particular role of strategies based on negative religious reappraisal.

**Author Contributions:** Conceptualization, N.Z. and E.G.; Formal analysis, N.Z.; Investigation, N.Z.; Methodology, N.Z., and K.B.-M.; Writing—original draft, N.Z. and K.B.-M.; Writing—review & editing, N.Z., K.B.-M. and E.G. All authors have read and agreed to the published version of the manuscript.

**Funding:** This research was funded by the statutory research grant of the Faculty of Psychology of SWPS University of Social Sciences and Humanities, Warsaw, Poland (no. WP/2018/B/40) and by the statutory research grant of the Faculty of Psychology of University of Warsaw, Warsaw, Poland.

**Conflicts of Interest:** The authors declare no conflict of interest. The funders had no role in the design of the study; in the collection, analyses, or interpretation of data; in the writing of the manuscript, or in the decision to publish the results.

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
