# Peer review of "Emotional State of Parents of Children Diagnosed with Cancer: Examining Religious and Meaning-Focused Coping"

_religions, doi:10.3390/rel11030132_

Round 1

Reviewer 1 Report

The paper titled “Emotional state of parents of children diagnosed with cancer: examining religious and meaning-focused coping” examines an interesting and important topic, the relationship between positive and negative religious coping and positive and negative emotions and cognitive reappraisal. It is a cross-sectional study, well designed. However, in its present form it has several shortcomings that need to be addressed.

The paper examines complex theoretical concepts which require precision in expression. The paper would benefit a lot if it was proofread by a person whose first language is English.

Introduction:

Most of the first paragraph of the introduction presents very general information that may be reduced substantially. The “space” could be used to cover the existing literature on meaning making, when faced with a stressor, and religious coping.

The third paragraph could be expanded so that the authors present systematically the findings/conclusions, if any, of the extant literature regarding the questions examined. In its present form this paragraph is confusing because, in my opinion the findings are not synthesized. Also, at some points the term religion is used, at other points the term religious coping. Is there a difference?

Lines 68-69 are inconsistent to line 80, I think. Initially, the two types of religious coping are presented as having different correlates but in the next paragraph they are presented as the opposite poles of one dimension.

The fourth paragraph lists the questions of the study but references concepts and relationships that are not addressed earlier.

Materials and Methods:

            The measurement of turning to religion, a type of religious coping, is introduced in this section but its differences from the other measures of religious coping, included in this study, are not addressed in the Introduction.

Results:

            My only concern in this section is the ratio of participants to independent variables for the regression analyses. It is far too low.

Discussion:

            This section is confusing to me. The authors have some interesting findings. Maybe if they had smaller paragraphs based on which they could organize their findings this whole section would be clearer. Also, in my opinion, the presentation of the existing literature at the lower half of page 9 belongs to the introduction section.

            Lines 252-253 “there is no concept of negative meaning finding”. I would suggest the authors consult the Park & Folkman (1997, pg. 130) paper.

Author Response

Thank you very much for your suggestions and remarks concerning our article entitled "Emotional state of parents of children diagnosed with cancer: examining religious and meaning-focused coping", which we would like to publish in Religions, Special Issue "Spiritual Care for People with Cancer".

Below we cite every remark and comment of the Reviewers and provide our answers to them in the parentheses.

All the corresponding changes in the manuscript are marked with red font.

Reviewer 1.

  • Reviewer 1 comments:

“The paper examines complex theoretical concepts which require precision in expression. The paper would benefit a lot if it was proofread by a person whose first language is English”.

Author’s response: Thank you very much for this comment. The revised version of the manuscript has been professionally proofread by a native speaker as suggested.

  • Reviewer 1 comments:

“Introduction: Most of the first paragraph of the introduction presents very general information that may be reduced substantially. The “space” could be used to cover the existing literature on meaning making, when faced with a stressor, and religious coping.

Authors’ response: Thank you very much. As this was also raised by other Reviewers, we have rewritten the whole introduction to more in-depth elaborate on the constructs. For details, please see the manuscript, lines 10-18, 20-29, 34-35, 58-67, 71-93.

The third paragraph could be expanded so that the authors present systematically the findings/conclusions, if any, of the extant literature regarding the questions examined. In its present form this paragraph is confusing because, in my opinion the findings are not synthesized. Also, at some points the term religion is used, at other points the term religious coping. Is there a difference?

 Authors’ response: Thank you very much for bring our attention to this lack of synthesis. We have also more precisely distinguished between religion and religious coping. The changes can be found in the manuscript, lines 48-51.

Lines 68-69 are inconsistent to line 80, I think. Initially, the two types of religious coping are presented as having different correlates but in the next paragraph they are presented as the opposite poles of one dimension.

Authors’ response: Thank you for this comment. We analyzed this study once again and we have rewritten results of this study, line: 62-67.

The fourth paragraph lists the questions of the study but references concepts and relationships that are not addressed earlier”.

Authors’ response: We have significantly changed the introductory part of the article.

We hope that after these changes, the introductory part provides a better justification for the questions of the study.

  • Reviewer 1 comments: Materials and Methods:

            The measurement of turning to religion, a type of religious coping, is introduced in this section but its differences from the other measures of religious coping, included in this study, are not addressed in the Introduction.

Authors’ response:  Thank you for this comment. The creators of the Brief Cope tool used various concepts and theoretical models addressing mainly Lazarus’s transactional theory of stress and behavioral self-regulation (Carver, Scheier, Weintraub, 1989). 

Due to the fact that this publication is an article, it is difficult to present all of the concepts in it. Therefore, we only signal which tool has been used and present a description of this scale we have utilized in our research. We added an information in lines 71-83.

  • Reviewer 1 comments:

“Results: My only concern in this section is the ratio of participants to independent variables for the regression analyses. It is far too low”.

Authors’ response:   We recognize that this is a limitation of our research.

The number of participants is the effect of the difficulties connected with the availability of the study group and the fact that the study is time-consuming. However, in future research it is worth to get a larger group size, to make the results even more precise.

  • Reviewer 1 comments:

“Discussion:

This section is confusing to me. The authors have some interesting findings. Maybe if they had smaller paragraphs based on which they could organize their findings this whole section would be clearer. Also, in my opinion, the presentation of the existing literature at the lower half of page 9 belongs to the introduction section.

Authors’ response: Thank you very much for this suggestion. We split paragraphs and we hope, that now it is clearer and more explicit. 

Lines 252-253 “there is no concept of negative meaning finding”. I would suggest the authors consult the Park & Folkman (1997, pg. 130) paper.”

Authors’ response: Thank you very much. As also other Reviewer pointed out our lack of precision here, we have rewritten this part to more accurately illustrate our line of reasoning, with reference to Park’s work on it (see lines 271-277). 

Reviewer 2 Report

I enjoyed this article and overall thought it was well-written. There were several minor grammatical errors and awkward English. Here are some examples:

Line 42 "sustaining and activate such emotions" (wording)

Line 48 "enables person" (wording)

Line 51 allow to restore resources (wording)

Line 98 the two main reaserch questions (typo)

Line 102  explain (typo)

Line 130 needs punctuation

Line 131-2  15 adjectives base on questionnaire

Line 174 any of the models did not reach the statisticial significance (wording and typo)

Line 202-203 the so-called objective characteristics of the situation has a very narrow explanatory power to differentiate the emotional response of the parent when sample is relatively (wording)

Line 257 misplaced comma

I am not an expert statistician. Although the statistical analysis seemed ok to me--I hope that the editors will get a quantitative researcher to check this research analysis. (I am a qualitative researcher).

In 2.3 when describing the measures used, the authors should add the reliability and validity measurements of the questionnaires/scales used in the study.

Author Response

Thank you very much for your positive opinion as well as very careful reading.  The extensive editing has been done on the revised version followed by professional proofreading. All the changes have been introduced unless the phrase has been totally rewritten due to comments by other Reviewers. Line 189-191, 194-196 has been rewritten.

  • Reviewer 2 comments:

“I am not an expert statistician. Although the statistical analysis seemed ok to me--I hope that the editors will get a quantitative researcher to check this research analysis. (I am a qualitative researcher).

In 2.3 when describing the measures used, the authors should add the reliability and validity measurements of the questionnaires/scales used in the study”

Authors’ response: Thank you very much for this suggestion. We have added relevant information about reliability and validity of the scales used in the study (lines: 139-141, 147-148, 152-154). The scales are widely used and have good psychometric characteristics. In our study reliability reported by Cronbach’s alpha was also satisfactory, with exception for positive reappraisal subscale, where this value equals .61.  The subscale consists only of two items, which may affect internal consistency (Carver, 1997). However, in other papers using this subscale its validity measured by the same indicator has usually been higher (Perczek, Carver, Price, Pozo-Kaderman, 2000), thus this lower value can be sample specific. A closer look at the content of the items (see below) suggests that the second item may have different meaning for parents of children with possibly fatal somatic condition. First, it may be an act of denial more than positive reframing, second, it may be perceived as not valid or even insensitive or cruel. Thus, the correlation between both items is only very moderate (.44). We

discussed it as a potential limitation but also valid information on our sample, line: 303-305.

  1. I've been trying to see it in a different light, to make it seem more positive.
  2. I've been looking for something good in what is happening.

Carver, C.S. (1997). You Want to Measure Coping but Your Protocol’s Too Long: Consider
            the Brief-COPE. International Journal of Behavioral Medicine, 4, 1, 92-100.

Perczek, R., Carver, C.S., Price, A.A., Pozo-Kaderman, C. (2000). Coping, Mood, and
            Aspects of Personality in Spanish Translation and Evidence of Convergence With
            English Versions. Journal of Personality Assessment, 74,1, 63-87.

Reviewer 3 Report

Topic and general thoughts

The paper focuses on important an important topic Emotional state of parents of children diagnosed with cancer: examining religious and meaning-focused coping. As said in the paper, there is a high need to better understand the phenomena. When taking into consideration of the topic and utilized theories, I was surprised that Lazarus’ Stress and Emotions was not referred to. Yet, I find more important to bind the study to discussions on meaning-making bit more thoroughly as now the paper is strongly linked to work of Pargament.  

I will next bring out few papers that I thought that are likely to resonate with what the authors have written. I am not saying that all of these papers need to be referred, I am saying that there are links to these that would bring the paper to the next level. There are many parts in the paper that resonates with the work of Crystal Park. As she has written a lot about cancer, coping and meaning-making, her work can be seen as relevant to the paper. E.g.

Park, C. L. (2013). Religion and meaning. In R. F. Paloutzian & C. L. Park (Eds.), Handbook of the psychology of religion and spirituality, 2nd Edition (pp. 357-379). New York: Guilford

Park, C. L. (2010). Making sense of the meaning literature: An integrative review of meaning making and its effects on adjustment to stressful life events. Psychological Bulletin, 136, 257–301.

Further there are things that could be discussed with Tatjana Schnell's work on Sources of meaning and meaning in life -studies. From Schnell’s work, e.g.

Schnell, T. (2009). Sources of meaning and meaning in life questionnaire (SoMe): Relations to demographics and well-being. The Journal of Positive Psychology, 4(6), 483−499.

Schnell, T. (2012). Spirituality with and without religion: Differential relationships with personality. Archive for Psychology of Religion, 34(1), 33-61.

Schnell, T., & Keenan, W. J. F. (2013). The construction of atheist spirituality: A survey-based Study. In H. Westerink (Ed.), Constructs of meaning and religious transformation: Current issues in the psychology of religion (pp. 101-119). Göttingen: V&R Unipress.

Style of writing

The text is mainly easy to follow. Yet, from the view point of academic English it would be beneficial to go through the text with the thought that one paragraph includes one idea/theme. Especially, introduction and discussion include lengthy paragraphs.

Context of the study

Context of the study is not explained. As American measurements are utilized, context of the study should be explained -> Many studies note that RCOPE is not fully transferable to the European context.  If this a study conducted in Europe, something should be said on lack of RCOPE.  I am not saying that RCOPE could not be utilized in Europe (I think it can and it is a valuable piece of work), but there needs to be an understanding how it works in this specific context (as majority of the participants were Catholic, something could be said on the general religiosity and religious participation in the country + connect the religious context to the usage of RCOPE). Further, it might be interesting to shortly clarify the definition of religion (as including spirituality) as in might leave some blind spots.
For the critique on RCOPE, see (most of these should be found from online free of charge) e.g.:

Stifoss-Hanssen, H. (1999). Religion and spirituality: What a European ear hears. The International Journal for the Psychology of Religion, 9(1), 25-33.

Xu, J. (2015). Pargament's theory of religious coping: Implications for

spiritually sensitive social work practice. British Journal of Social Work, 46(5), 1394-1410.

Kwilecki, S. (2004). Religion and coping: A contribution from religious

studies. Journal for the Scientific Study of Religion, 43(4), 477-489.

Alma, H.A., Pieper, J.Z.T. & Uden, M.H.F. van (2003). When I find myself in times of trouble: Pargament’s religious coping scales in the Netherlands. Archive for the Psychology of Religion, 24, 64-74.

Discussion

251-2 “In contrast, negative religious coping does not match this construct because there is no concept of "negative" meaning finding.” This sentence points out the need to dig in deeper with Park and Schnell. If I follow the argumentation, you are now talking about loss of meaning and there is a lots of studies on that.

Author Response

Thank you very much. Indeed, we should have elaborated more deeply on issues you pointed out. We have rewritten the whole introduction to include all the suggested threads, with special focus on Park’s work on meaning and Lazarus’s work on emotions. We have done more extensive literature review and included all the papers indicated as relevant and among. The details can be found in manuscript, especially in lines 20-29, 84-93.

  • Reviewer 3 comments:

“Style of writing

The text is mainly easy to follow. Yet, from the view point of academic English it would be beneficial to go through the text with the thought that one paragraph includes one idea/theme. Especially, introduction and discussion include lengthy paragraphs.”

Authors’ response:  Thank you very much for this practical suggestion. The text has been extensively edited and corrected in this regard.

  • Reviewer 3 comments:

“Context of the study

Context of the study is not explained. As American measurements are utilized, context of the study should be explained -> Many studies note that RCOPE is not fully transferable to the European context.  If this a study conducted in Europe, something should be said on lack of RCOPE.  I am not saying that RCOPE could not be utilized in Europe (I think it can and it is a valuable piece of work), but there needs to be an understanding how it works in this specific context (as majority of the participants were Catholic, something could be said on the general religiosity and religious participation in the country + connect the religious context to the usage of RCOPE). Further, it might be interesting to shortly clarify the definition of religion (as including spirituality) as in might leave some blind spots.

For the critique on RCOPE, see (most of these should be found from online free of charge) e.g.:

Stifoss-Hanssen, H. (1999). Religion and spirituality: What a European ear hears. The International Journal for the Psychology of Religion, 9(1), 25-33.

Xu, J. (2015). Pargament's theory of religious coping: Implications for spiritually sensitive social work practice. British Journal of Social Work, 46(5), 1394-1410.

Kwilecki, S. (2004). Religion and coping: A contribution from religious studies. Journal for the Scientific Study of Religion, 43(4), 477-489.

Alma, H.A., Pieper, J.Z.T. & Uden, M.H.F. van (2003). When I find myself in times of trouble: Pargament’s religious coping scales in the Netherlands. Archive for the Psychology of Religion, 24, 64-74.”

Authors’ response: Thank you very much for raising this important issue. It is a broader discussion about emic and etic validity. As you have already mentioned, RCOPE is a tool successfully used in studies among very diversified populations (Abu-Raiya, Pargament, 2015). Please notice that in the later works the questionnaire has been generally accepted to be valid among Christians worldwide, but also (with minor modifications or without) among people who declared different religion, even outside the Christian culture circle (Abu-Raiya, et al., 2011, Gardner et al., 2013). Additionally, people who are non-religious or religious but with dogmas other than personal God may choose the answer that they do not apply these strategies at all.  However, we entirely agree that as every self-descriptive tool created in a given sociocultural context, RCOPE has its limitations. For instance, Xu (2015) highlighted that without qualitative approach the phenomenon of religious coping is neither fully understood nor covered, including unconscious ways and mechanisms of religious coping (Kwilecki, 2004). 

Responding to this comment with regard to the specific context of our study, it was conducted in two medical centers in Poland. According to the Central Statistical Office (2018), 94% of Poles declare themselves as members of religious denomination, with 92% as members of the Roman Catholic Church. Additionally, 81% of those surveyed declared themselves as believing in God. This shows that our sample was derived from very coherent population in this regard (comparing to other countries in Europe) with religious background based on personal God who can be perceived as a source of support or punishment and guilt, which is the essence of RCOPE positive and negative religious coping, respectively. Thus, these constructs are present in the culture and as such are experienced in everyday life by a layman so it can be assumed that they have sufficient emic validity.

Also, in our study 91,1% of the parents self-declared as being religious and during both ethical evaluation as well pilot study no questions were raised in relation to items’ content and tool validity. Moreover, RCOPE has an official Polish adaptation and has been used in many studies without any reports of its poor fit from emic perspective. However, the results we obtained cannot be generalized not only to different cultural contexts, but even to different samples (e.g., parents of children suffering from other diseases) within the same cultural context.

We have just added the shorter version of this elaboration in the description of the tool to allow the readers to formulate their own opinion on the limitations of the findings of our study (see lines 132-134).

Abu-Raiya, H., Pargament, K.I. (2015). Religious Coping Among Diverse Religions: Commonalities and Divergences. Psychology of Religion and Spirituality, 7,1, 24-33.

   Journal for the Scientific Study of Religion, 43(4), 477-489.

Abu-Raiya, H., Pargament, K. I., & Mahoney, A. (2011). Examining coping methods with
            stressful interpersonal events experienced by Muslims living in the United States
            following the 9/11 Attacks. Psychology of Religion and Spirituality, 3, 1–14.

Gardner, T. M., Krägeloh, C. U., & Henning, M. A. (2013). Religious coping, stress, and
            quality of life of Muslim university students in New Zealand. Mental Health, Religion
            & Culture
. Advance online publication.

Kwilecki, S. (2004). Religion and coping: A contribution from religious studies. Journal for
            the Scientific Study of Religion, 43(4), 477-489.

Xu, J. (2015). Pargament's theory of religious coping: Implications for spiritually sensitive social work practice. British Journal of Social Work, 46(5), 1394-1410.

  • Reviewer 3 comments:

“Discussion: 251-2 “In contrast, negative religious coping does not match this construct because there is no concept of "negative" meaning finding.” This sentence points out the need to dig in deeper with Park and Schnell. If I follow the argumentation, you are now talking about loss of meaning and there is a lot of studies on that”.

Authors’ response: Thank you very much. We have formulated this line of reasoning more precisely after careful reading of Park and Schnell as suggested. For the current version please look at the manuscript, lines 271-277.

Finally, we would like to thank the Editor and Reviewers once again for their suggestions and remarks concerning our manuscript. We found all the comments very useful as they provided us with invaluable help and guidance on how to improve the manuscript quality. We highly appreciate the chance to revise our manuscript and re-submit it to the special issue of Religions.

Round 2

Reviewer 1 Report

The paper titled “Emotional state of parents of children diagnosed with cancer: examining religious and meaning-focused coping” is much improved. However, I still have some concerns and questions.

I still think the paper needs editing for typos and language errors.

Introduction:

I would delete the 2nd paragraph.

Discussion:

            This section is still confusing to me. I have difficulty reconciling the text with the Tables in paragraphs 2 & 3. Are they consistent? Are there typos?

Reviewer 3 Report

I thank the authors for engaging in the extensive writing process and providing such detailed comments to the review report. All the issues I raised in my report, are now fixed.

Only a few small remarks:

146 tuning to religion – turning?

215 valences or variances?

311 comma is blue (not black)
